# Realistic prediction and engineering of high-Q modes to implement stable Fano resonances in acoustic devices

Felix Kronowetter [1,2,3] ✉, Marcus Maeder [1], Yan Kei Chiang [2], Lujun Huang [2], Johannes D. Schmid [1], Sebastian Oberst [3], David A. Powell [2] & Steffen Marburg[1]

Quasi-bound states in the continuum (QBICs) coupling into the propagating spectrum manifest themselves as high-quality factor (Q) modes susceptible to perturbations. This poses a challenge in predicting stable Fano resonances for realistic applications. Besides, where and when the maximum field enhancement occurs in real acoustic devices remains elusive. In this work, we theoretically predict and experimentally demonstrate the existence of a Friedrich-Wintgen BIC in an open acoustic cavity. We provide direct evidence for a QBIC by mapping the pressure field inside the cavity using a Laser Doppler Vibrometer (LDV), which provides the missing field enhancement data. Furthermore, we design a symmetry-reduced BIC and achieve field enhancement by a factor of about three compared to the original cavity. LDV measurements are a promising technique for obtaining high-Q modes' missing field enhancement data. The presented results facilitate the future applications of BICs in acoustics as high-intensity sound sources, filters, and sensors.

Bound states in the continuum were first established by Neumann and Wigner[1] in the context of an electron that remains in its orbit, although it has enough energy to overcome the attractive forces and propagate to infinity. The transfer to acoustics was made by Ursell[2] followed by seminal works on symmetry-protected BICs[3–6], Fabry–Pérot BICs[7–9], and Friedrich–Wintgen BICs[8,10–12].

In linear acoustics, the continuous spectrum of an open system is spanned by propagating waves that radiate to infinity, i.e., the solutions corresponding to complex eigenfrequencies greater than or equal to the cut-off frequency of the system. Propagating waves can also be described as extended states with an outgoing energy flux. Apart from the continuum or several continua, the total frequency spectrum of the open system includes localized solutions corresponding to discrete eigenvalues. Localized solutions corresponding to discrete eigenvalues outside the continuous spectrum are called bound states. Bound states are perfectly confined waves since they are completely decoupled from open propagation channels and carry no outgoing energy flux, hence they can't radiate away. In contrast, localized solutions in the continuum generally couple to open propagation channels, becoming leaky resonances. The eigenfrequency of the highly localized quasi-trapped modes is complex, in which the real part denotes the resonance frequency, and the imaginary part characterizes the radiation loss[13,14]. For a particular configuration of the geometric parameters, the radiation loss vanishes and the resonances become confined states. These localized solutions to discrete eigenvalues coexisting with propagating waves are known as embedded trapped modes or, more commonly BICs. From the mathematical point of view, the discrete eigenvalues of BICs can be described by purely real eigenfrequencies. Since BICs are localized solutions, they are invisible to extended states in the first propagation channel, hence they cannot be excited by them.

[1]Chair of Vibro-Acoustics of Vehicles and Machines, Department of Engineering Physics and Computation, TUM School of Engineering and Design, Technical University of Munich, Bavaria, Germany. [2]School of Engineering and Information Technology, University of New South Wales, Northcott Drive, Canberra, ACT 2600, Australia. [3]School of Mechanical and Mechatronic Engineering, Centre for Audio, Acoustics and Vibration, Faculty of Engineering and IT, University of Technology Sydney, Sydney, Australia. ✉e-mail: felix.kronowetter@tum.de

In recent years, extensive theoretical studies on numerous BIC configurations and BIC tuning have been conducted[15–23]. Additionally, geometrical phase engineering of BICs extending Fano resonances beyond their conventional limits is investigated[17]. Furthermore, the theoretical and experimental demonstration of BIC-induced high Q-factors[24,25] can be applied to the design of novel, high-performance acoustics sensors. Huang et al.[26] demonstrate a Friedrich–Wintgen quasi-BIC leading to emission enhancement of a sound source by nearly two orders of magnitude. Recent studies have suggested that acoustodynamic devices can be used for quantum computing[27–30], where the application of mechanical BIC-induced high-Q Fano resonances (e.g., Yu et al.[31]) could be of further interest. A comprehensive review of BICs can be found in Hsu et al.[32], Pagneux[33], Joseph[34], and Sadreev[35].

All aforementioned studies lack information on the exact sound pressure field inside the resonant cavity leading to BIC formation under realistic conditions. Here, we demonstrate a BIC induced by mode interference or Friedrich–Wintgen BIC. The specific design of the resonator geometry and measurement technique allows us to measure the transmission spectrum and map the sound pressure field inside the cavity using laser Doppler vibrometry. This technique leads to the first visual proof of an acoustic BIC and, most notably, provides direct access to the pressure distribution inside the cavity. Thus, it is possible to run a detailed analysis of the sound pressure field and its contribution to the reflection or transmission spectrum. Our findings lead to a new type of Friedrich–Wintgen BIC relying on reduced symmetry and the principle of mirror sources resulting in a high-Q mode without exciting unwanted resonances. We further present the first direct visualization of a Friedrich–Wintgen BIC using laser Doppler vibrometry as a pressure field mapping technique. We use laser Doppler vibrometry measurements to obtain a complete mapping of the sound pressure field to better understand BIC formation in the presence of realistic losses. The reflection and transmission spectra are obtained using microphones but do not provide information about the exact pressure enhancement inside the cavity. The pressure distribution inside the cavity is needed to develop high-performance acoustic devices based on BICs, such as acoustic sources and sound lasers. Because BICs are extremely sensitive, any perturbation of the high-Q mode, energy extraction, or backscattering from microphones will degrade the BIC, so we use this technique to avoid any perturbation of

the pressure field. By mapping the sound pressure field of the BIC, we have direct access to the actual pressure values and thus to the critical information of when and where the pressure magnitude reaches a maximum. The interaction between the localized thermo-viscous losses and the concentrated high-intensity fields of the BIC is the key to determining the achievable Q-factor and pressure field enhancement. We facilitate the accurate analysis of high-Q modes in the presence of realistic losses, determine the configuration with maximum pressure enhancement, and enable the future application of high-Q Fano resonances to acoustic devices.

## Results

Here, we consider a BIC associated with the Friedrich–Wintgen full destructive interference of degenerate modes of the same symmetry. Friedrich and Wintgen[10] demonstrated the formation of BICs due to the interference of resonances belonging to different channels. In contrast to symmetry-protected BICs in waveguides, Friedrich-Wintgen BICs occur above the cut-off frequency of the first duct cross-mode (antisymmetric about the duct axis) and close to the point of modal degeneracy of the closed system. Friedrich-Wintgen BICs have the special feature that BICs still form, even if the symmetry of the system is broken[8]. An open system with a non-Hermitian Hamiltonian is a prerequisite for the observation of BICs since they are forbidden in compact systems[36,37]. We use coupled-mode theory[15,23,35,38] to predict the point of modal degeneracy for a closed cavity. See Supplementary Information Section 1 for a complete theoretical analysis. By coupling the cavity to an acoustic waveguide, the localized solutions of the closed cavity turn into leaky resonances. Hence, we investigate the complex eigenfrequencies and corresponding modes of a resonant cavity with open ends. By variation of the geometrical parameters, the relevant eigenfrequencies and modes are identified, forming a BIC. The existence of a Friedrich-Wintgen BIC is numerically shown for a non-rotationally symmetric duct-cavity structure with open ends, i.e., a rectangular cavity placed in a tubular waveguide. A schematic illustration of the structure is displayed in Fig. 1a.

We chose this structure since the formed BIC is stable against asymmetry and isolated in the studied frequency range, i.e., the BIC is robust even in the presence of manufacturing imperfections and no further resonance peaks are found. Embedded trapped modes can only be found for a particular configuration of the geometric

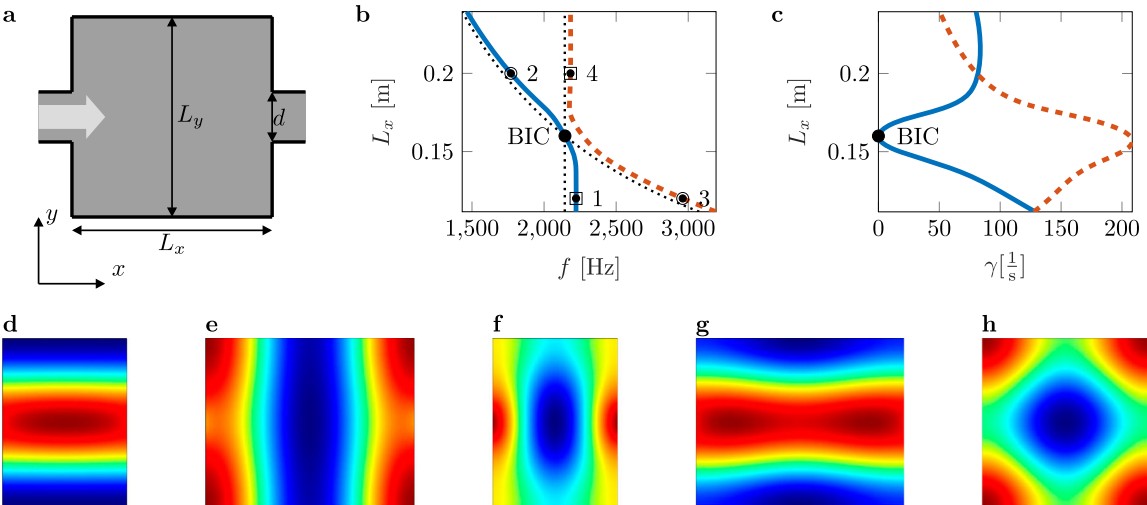

**Fig. 1 | BIC in an open acoustic resonator. a** Schematic of a resonant cavity coupled to an acoustic waveguide. The z-axis is perpendicular to the xy-plane. The length $L_y$ is set to 160 mm and the diameter $d$ of the pipe to 40 mm. **b** Avoided crossing of the real parts of high-Q and low-Q modes for varying length $L_x$ of the resonant cavity. **c** Vanishing imaginary part of the BIC, where the BIC mode

dominates the decay process. **d**–**h** Interacting modes of different configurations of $L_x$ (**d** $M_{131}$ (at point 1 in (**b**)), **e** $M_{311}$ (at point 2 in (**b**)), **f** $M_{311}$ (at point 3 in (**b**)), **g** $M_{131}$ (at point 4 in (**b**))), as well as the mode shape of the BIC (**h** $M_{331}$), are shown, corresponding to the points marked in (**b**).

parameters of the system and are a result of modal coupling via a common continuum, i.e., the interaction of modes of the same symmetry. As a consequence of the interaction, avoided crossings are observed, where the real parts of the eigenfrequencies cross with modal interchange and a weakly damped resonance dominates the decay process. This is also called resonance trapping[39,40] and is shown by the blue and orange lines in Fig. 1b, c. The exact mechanism of BIC formation as well as the complex eigenfrequencies are presented in Supplementary Information Sections 2 and 3. The thin black dotted lines are the results of the coupled mode theory for a closed cavity. Consequently, the crossing point of the black lines is the point of modal degeneracy and exactly predicts the BIC. We solve a numerical problem, including thermo-viscous losses, to take dissipation in the boundary layer into account, leading to a reduced Q-factor (shown in Supplementary Information Section 4). Referring to Lyapina et al.[15], the leaky modes are denoted by $M_{pqr}$, with $p$, $q$, and $r$ being the number of maxima in the pressure field along the $x$-, $y$-, and $z$-axis, respectively. The distance $L_x$ is varied with the resulting modal interchange depicted regarding mode $M_{131}$ in Fig. 1d, becoming mode $M_{311}$ in Fig. 1e and mode $M_{311}$ in Fig. 1f, turning into mode $M_{131}$ in Fig. 1g, respectively. Additionally, the BIC mode $M_{331}$ is shown in Fig. 1h. The BIC is observed at a frequency of $f = 2145\,Hz$ with a cavity length of $L_x = 160\,mm$. Excellent agreement is found between the numerically computed eigenfield profile of the Friedrich–Wintgen BIC and the eigenfield profile predicted by the analytical solution, see Supplementary Information Section 1.

## Experimental verification of BIC

To experimentally demonstrate the existence of the predicted BIC, we manufacture three samples of the resonant cavity with varying dimensions $L_x$ shown in Fig. 2a.

The samples are fabricated using selective laser sintering to keep manufacturing tolerances small and avoid asymmetry in our geometry. Furthermore, the walls are designed to be thick enough to suppress structural resonances in the frequency range of interest. In experiments, we use an impedance tube to obtain the transmission spectrum. The measurement setup is depicted in Fig. 2b, where the sample is integrated into the tube. The diameter of the tube, as well as the

height and depth of the cavity, are kept constant at $d = 40\,mm$, $L_y = 160\,mm$, and $L_z = d = 40\,mm$, cf. Fig. 1a. The length of the cavity $L_x$ is set to 160, 165, and 170 mm (left to right in Fig. 2a), respectively. Hence, we realize a parameter variation in the vicinity of the BIC configuration.

BICs couple into extended states and become QBICs if the BIC configuration of the system is disturbed. QBICs are slightly damped complex resonances that radiate energy and reveal themselves in the form of Fano resonances. Fano resonances are a well-studied phenomenon in many different fields of physics, e.g., see the detailed review by Miroshnichenko et al.[41]. Therefore, the second and third samples are designed such that QBICs can be measured. Due to the destructive interference of bound states and continua, the typical resonance and antiresonance features of the asymmetric Fano resonances can be observed, in the sound transmission spectrum, see Fig. 2c. In contrast to the BIC, we identify Fano peaks of finite height and increased width.

The Fano peak widens and a frequency shift toward lower frequencies is observed due to the increased cavity volume as $L_x$ is increased from the BIC configuration. We can see that the numerical results coincide with the experimental ones and also show the expected behavior. Additionally, the transmission loss (TL) of the system is plotted over the frequency for all three configurations. The Fano peak of the transmission spectrum leads to a high amplitude in the TL. Similar to the previously mentioned behavior of the Fano peaks, the TL peaks broaden and lessen with increasing $L_x$. Nevertheless, the TL peak of $L_x = 165\,mm$ seems to be lower than the one of $L_x = 170\,mm$ shown in Fig. 2d. Thermo-viscous losses affect the amplitude of the TL more strongly the closer we get to the BIC configuration. The measured Q-factors for $L_x = 165\,mm$ and $L_x = 170\,mm$ are 328 and 302, respectively. In the case of the BIC configuration being restored, the Fano resonances collapse. This state is described as the ghost of the Fano resonance by Ladron de Guevara et al.[42] and can be identified in Fig. 2c, d regarding the blue lines, i.e., no Fano peak is present.

## Visualization of QBIC

Three continua exist in our duct-cavity structure. The first continuum is symmetric to the duct axis, with a lower limit defined by the cut-off

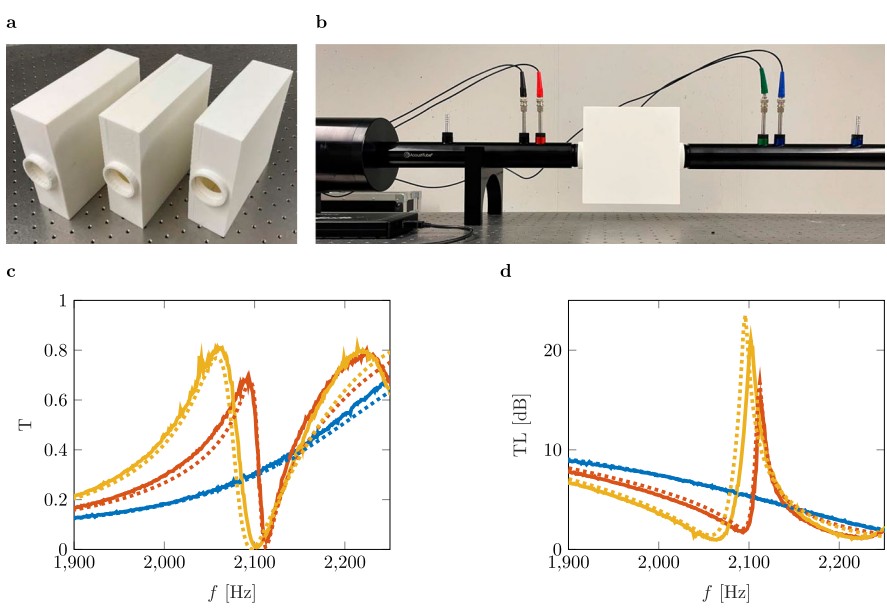

**Fig. 2 | Transmission spectra of cavities of varying lengths. a** Images of three manufactured samples with different dimensions $L_x$. **b** Transmission measurement set-up. **c, d** Transmission coefficient as well as the transmission loss in the frequency range 1900–2250 Hz. The blue lines represent the results of the measurement and the numerical simulation of the BIC configuration ($L_x = 160\,mm$), respectively. The orange lines represent the results of $L_x = 165\,mm$, and the yellow lines are the ones of $L_x = 170\,mm$.

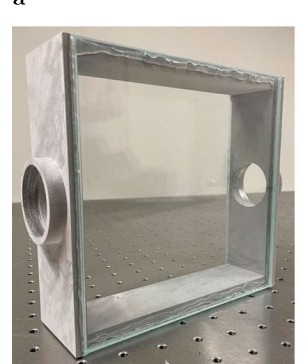
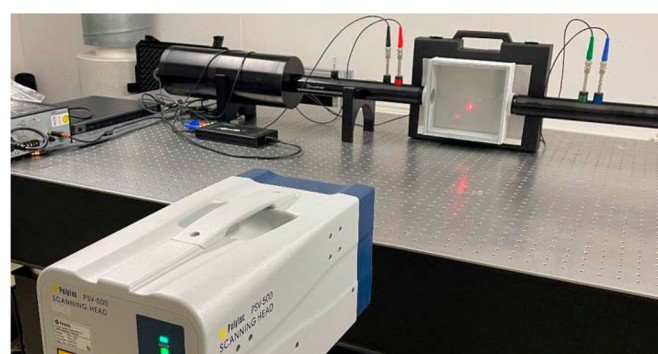

**Fig. 3 | Experimental set-up for the visualization of the sound pressure field. a** Printed sample with high-transmission glass mounted as side walls. **b** Experimental set-up for the refracto-vibrometry measurements.

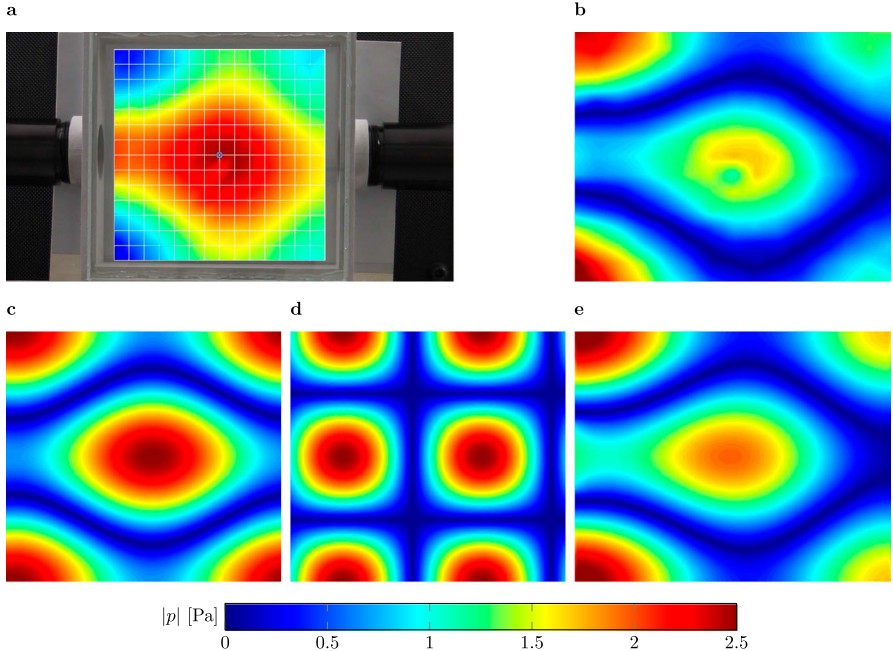

$|p|$ [Pa]

| 0 | 0.5 | 1 | 1.5 | 2 | 2.5 |

**Fig. 4 | Sound pressure field visualization. a** Visualization of the real part of the sound pressure field inside the cavity. The red color indicates a pressure maximum and the blue color a minimum, respectively. **b** Absolute value of the measured pressure field. **c** Superposition of the four most dominant modes (modes with the highest FFT coefficients). **d** Superposition of the next four modes with high coefficients. **e** Simulation results. The colored scales display the absolute values of the pressure in Pa, with the pressure being normalized to the incident pressure field.

frequency of the duct. The second and third continua have lower limits defined by the cut-off frequencies of the first duct cross-mode and the first cavity cross-mode (symmetric to the $y$-axis), respectively. The shortest side of the cavity is chosen such that it matches the diameter of the pipe. Therefore, we expect unitary pressure distribution in this direction. This enables experiments applying refracto-vibrometry to visualize the sound pressure field of the QBIC. A new sample is manufactured, allowing a laser to pass through the structure. To do so, two side walls are replaced by high-transmission glass, as shown in Fig. 3a.

We chose the $L_x = 170$ mm configuration of the previous measurements for the new sample since the Fano peak is the widest. This is crucial for the measurement, in which we have a single excitation frequency matching the frequency of the QBIC. Due to the sensitivity of the QBIC, the gradient of the Fano peak should be as low as possible to ensure that we find the QBIC. The sample is placed in the impedance tube set-up.

To conduct the refracto-vibrometry, a laser Doppler scanning vibrometer PSV-500 from Polytec (Polytec Gmbh, Waldbronn,

Germany) is used to measure the changes of the refractive index of the fluid, which is proportional to the acoustic pressure variation within the cavity[43–46]:

$$v(\omega) = \omega \frac{1}{\gamma p_0} \frac{n_0 - 1}{n_0} \int_L p(l,\omega)\,\mathrm{d}l \qquad (1)$$

where $\omega$ is the angular frequency, $v$ is the LDV velocity, $p$ is the sound pressure, $n_0$ is the refractive index of air at standard atmospheric pressure, $p_0$ is the static atmospheric pressure, and $\gamma$ is the specific heat capacity ratio of air. The basic principle of the LDV is based on the well-known Doppler shift. The pressure waves inside the cavity slightly shift the phase of the emitted monochromatic laser light. The superimposition of the reflected and emitted laser light produces a speckle pattern on the photodetector, which allows the measurement of the corresponding frequency of the pressure waves and the change in the refractive index. The latter is proportional to the sound pressure inside the cavity. This makes it possible to visualize the corresponding

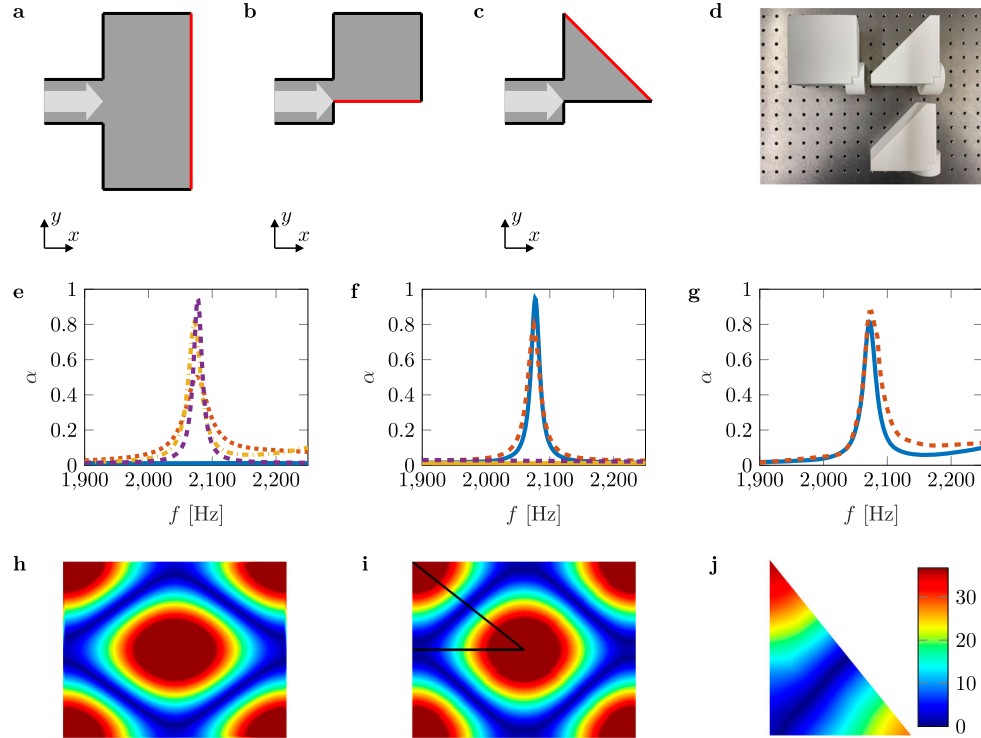

**Fig. 5 | Experimental verification of symmetry-reduced BICs. a** Schematic of a resonant cavity similar to that shown in Fig. 1a. The $z$-axis is perpendicular to the $xy$-plane. The length $L_y$ is set to 160 mm, and the diameter $d$ of the pipe to 40 mm. We halve the cavity at the axis of symmetry (highlighted in red) based on the principle of mirror sources and thus obtain a cavity length of $L_x = 85$ mm, which corresponds to half the length of the configuration with $L_x = 170$ mm. **b** Further reduction of the configuration in a using the principle of mirror sources once again. **c** Fully reduced configuration. **d** Manufactured samples for configurations **b** ($L_x = 85$ mm) and **c** ($L_x = 85$ mm for the top sample and $L_x = 77$ mm for the bottom sample). **e** Absorption coefficient $\alpha$ in the frequency range 1900–2250 Hz. The blue line represents the result of the numerical simulation of the BIC configuration of **c** with $L_x = 77$ mm. The orange dotted line, the yellow dash-dotted line, and the purple dashed line represent the numerical results of $L_x = 85$ mm for the configurations

(**a**–**c**), respectively. **f** Absorption coefficient $\alpha$ in the frequency range 1900–2250 Hz. The blue and yellow lines represent the results of the numerical simulation of the BIC configuration of **c** with $L_x = 85$ mm and $L_x = 77$ mm, respectively. The orange and the purple dashed lines represent the measurement results of configuration **c** with $L_x = 85$ mm and $L_x = 77$ mm. **g** The computed and measured absorption coefficients for configuration **b** are represented by the blue and orange dashed lines, respectively. **h** Sound pressure field of the BIC mode inside the cavity with $L_x = 170$ mm. **i** Sound pressure field of the BIC mode inside the fully reduced cavity with $L_x = 85$ mm for configuration (**c**), framed by the black lines and extrapolated to the rectangular cavity. **j** Measured absolute sound pressure inside the fully reduced cavity with $L_x = 67$ mm excited at 2315 Hz. The color scale represents the absolute pressure in Pa.

pressure distribution. The measured pressure distribution of the QBIC is displayed together with the results of the corresponding finite element simulations in Fig. 4.

The visualized sound pressure field of the QBIC within the experimental set-up is presented in Fig. 4a. In addition, we show a mapping of the absolute values of the pure measurement data in Fig. 4b for better comparison with the numerical data displayed in Fig. 4e. We observe that the measurement is in good agreement with the numerical prediction. As expected, the pressure maxima are located at both the edges and the center of the cavity. Minor inaccuracies within the numerical prediction are due to real losses and uncertainties within the experiments, i.e., imprecise alignment of the parts or imperfect plane wave excitation. We note only a slight pressure fluctuation in the center, just below the marked node in Fig. 4a. The position of the LDV for this measurement point is normal incidence to the glass. This results in a lower signal-to-noise ratio, which leads to higher measurement errors. We obtain a stable image of the QBIC that is in good agreement with the simulated sound pressure field. Thus, we present the visual evidence of a QBIC.

The visualized field allows us to further analyze the pressure field. For this purpose, we apply a fast Fourier transform (FFT) on the BIC mode, which is the same as applying an overlap integral with modes of the closed cavity. The values of the Fourier coefficients indicate the contribution of the modes. We take the first four dominant modes that are degenerate with the $(0, 1)$ mode (for mode indexing, see

Supplementary Information Section 5). Thus, we consequently reassemble the modes by conducting an inverse FFT. The result is depicted in Fig. 4c. By further analyzing the FFT coefficients, we identify the modes being excited alongside the BIC mode. The modes with the next higher FFT coefficients are the modes that are degenerate with the $(1, 1)$ mode. Figure 4d shows the result of their modal superposition. This mode is slightly shifted due to the incoming waves from the sound source and can also be found by conducting a modal analysis. We observe this mode due to an anti-symmetric excitation because we place our sound source on one side of the cavity. We, therefore, deduce, being consistent with the coupled mode theory[15,23,35,38], that we can create a perfect BIC mode by eliminating the contribution of other modes. This is done by applying the principle of mirror sources, where we retain the properties of the full cavity but suppress all anti-symmetric modes, as shown in Fig. 5.

Figure 5a–c shows three schematics of the previously studied resonant cavities reduced in size by applying the principle of mirror sources. We cut the geometry in half at a particular axis of symmetry (marked by the red lines in the schematics mentioned above). The axis of symmetry has to be chosen with care since the BIC mode needs to be sustained, and all anti-symmetric modes with respect to this axis need to be suppressed. Figure 5c depicts the schematic of the fully reduced geometry. We manufacture three additional samples for impedance tube measurements, shown in Fig. 5d. Figure 5e shows the numerically obtained absorption coefficients of the configurations in Fig 5a–c with

$L_x = 85$ mm and also of the BIC configuration in Fig. 5c determined at $L_x = 77$ mm. The more we reduce the geometry, the more unwanted modes are removed. Accordingly, we observe a clear peak in the absorption spectrum. We perform impedance tube measurements to validate our observations. The measured absorption coefficients are presented in Fig. 5f for the fully reduced configuration, see Fig. 5c, and in Fig. 5g for the configuration shown in Fig. 5b. The measurement results agree with our prediction. We observe the collapse of the Fano resonance (yellow and dashed purple lines in Fig. 5f), the sharp peak of the Fano resonance with no further absorption in the spectrum shown (blue and dashed red lines), and thus prove the existence of a Friedrich-Wintgen BIC based on the concept of fully reduced symmetry without exciting unwanted modes. Figure 5h shows the unexcited QBIC mode for a rectangular cavity with $L_x = 170$ mm. The sound pressure field of the fully reduced geometry framed by the black lines is depicted in Fig. 5i. For comparison, we extrapolate the reduced sound pressure field onto the rectangular cavity. We infer from comparing Fig. 5h, i that identical sound pressure distributions are obtained. Finally, we manage to preserve the BIC mode although the geometry is reduced significantly and thus present a new type of fully reduced high-Q mode without exciting unwanted modes.

### Pressure enhancement

Impedance tube studies allow us to extract values for reflection spectra and absorption but not for cavity pressure enhancement. The missing cavity pressure information is provided by the LDV. The numerical and experimental procedure to determine the maximum pressure is presented in detail in Supplementary Information Section 6. Figure 5j shows the measured absolute pressure field inside the fully reduced cavity with $L_x = 67$ mm excited at 2315 Hz. This configuration leads to the maximum pressure enhancement possible for this structure with a measured peak value of 36.74 Pa. We additionally measure the sound pressure fields of the fully reduced cavity with $L_x = 64$ mm and $L_x = 70$ mm to prove the existence of a pressure peak. The measurements show peak pressures of 26.08 Pa and 32.26 Pa, respectively. To demonstrate the magnitude of the pressure enhancement, we measure the pressure field inside the full cavity with $L_x = 170$ mm for several frequencies. The maximum pressure is 12.61 Pa. Thus, the fully reduced cavity leads to the highest pressure enhancement of the investigated Friedrich–Wintgen BIC by a factor of about three compared to the full cavity.

Finally, we extract the sound pressure field of a QBIC mode. The sound pressure field inside the cavity gives us accurate information about the influence of losses and hence the stability and confinement of the mode. We also show that LDV measurements are a powerful tool for predicting the maximum pressure enhancement of high-Q modes, can resolve even small deviations from numerical predictions, and thus provide seminal guidance for the application of QBICs. Thus, we present the realistic sound pressure field of a stable high-Q mode, enabling further analysis and its application to acoustic devices.

## Discussion

We report the theoretical design, computation, experimental verification, and visualization of an acoustic Friedrich-Wintgen BIC in an open rectangular cavity. This is not only the first visual proof of an acoustic BIC but, above all, enables direct access to the pressure values inside the cavity. An exact analysis of stable high-Q modes facilitates its application to acoustic devices.

For this purpose, we design and manufacture three versions of the cavity with varying lengths. One to match the BIC configuration and two more to broaden and stabilize the Fano peak in the transmission spectrum. For an accurate prediction of the BIC, thermo-viscous losses are taken into account, leading to reduced peaks in the sound transmission spectrum. We find that the numerical results match the experimental ones accurately.

Furthermore, we manufacture a sample with high-transmission glass side walls to facilitate experiments applying refracto-vibrometry, and pioneer the direct visualization of an acoustic QBIC. Consequently, we have direct access to the pressure information of the high-Q modes inside the cavity. Exact mapping of the pressure field gives us a better understanding of real QBICs, including losses of all kinds, and hence enables further analyses of the excited modes.

We decompose the pressure field and identify higher-order modes excited simultaneously with the BIC mode. In addition, we further adapt the concept of designing BICs proposed by Huang et al.[23] and thus reduce the resonant cavity to the smallest possible size. As a result, we design a new type of BIC relying on fully reduced symmetry. Hence, we only excite the QBIC mode and suppress all unwanted modes. We verify our predictions by impedance tube measurements. Finally, we determine the configuration with maximum pressure enhancement by mapping the pressure fields of the fully reduced cavity with varying lengths. The fully reduced cavity leads to the highest pressure enhancement of the investigated Friedrich–Wintgen BIC by a factor of about three compared to the full cavity.

Our findings are a fundamental contribution to the study of BICs and open up entirely new opportunities in this field of research. Recent studies demonstrate an emission enhancement based on acoustic BICs. Nevertheless, prior to this work, it has not been shown when the large field enhancement happens if thermo-viscous losses are considered in real acoustic devices. Mapping the pressure field of the BIC is a promising technique to obtain the missing data and hence facilitates the application of BICs to high-intensity sound sources, acoustic devices, and nonlinear acoustics.

## Methods

### Analytical model

The eigenfield profile of the Friedrich-Wintgen BIC can be predicted from the following equation

$$\psi_{BIC}(x,y) \approx \cos\theta\,\psi_{31}(x,y) + \sin\theta\,\psi_{13}(x,y) \tag{2}$$

where

$$\cos\theta = \frac{A}{\sqrt{A^2+B^2}}, \quad \sin\theta = \frac{B}{\sqrt{A^2+B^2}}, \tag{3}$$

$$A = -\sqrt{\frac{L_y}{2\pi^2 L_x}}\left[\sin\left(\frac{\pi(L_y+1)}{L_y}\right) - \sin\left(\frac{\pi(L_y-1)}{L_y}\right)\right], \tag{4}$$

$$B = \sqrt{\frac{2}{L_x L_y}}. \tag{5}$$

The corresponding modes $\psi$ are defined by

$$\psi_{m,n} = \sqrt{\frac{\left(2-\delta_m^1\right)\left(2-\delta_n^1\right)}{L_x L_y}}\cos\left(\frac{\pi(m-1)(2x+L_x)}{2L_x}\right)\cos\left(\frac{\pi(n-1)(2y+L_y)}{2L_y}\right) \tag{6}$$

with $\delta_n^1$ and $\delta_m^1$ being the Kronecker delta. See Supplementary Information Section 1 for a complete theoretical analysis.

### Numerical simulations

All Simulations in this article are performed with the commercial software COMSOL Multiphysics (Acoustics Module). The speed of sound and the air density is $c_0 = 343$ m/s and $\rho_0 = 1.2$ kg/m³, respectively. We consider the walls of the cavity as well as the walls of the

waveguide to be rigid and hence apply sound hard boundary conditions. Additionally, we consider thermo-viscous losses in our system. We apply the no-slip condition for the velocity field and an isothermal condition for the temperature at the walls of the cavity. To ensure non-reflective boundary conditions at the ends of the waveguide, we apply perfectly matched layers. We perform modal analyses to compute the eigenvalues and corresponding modes and time-harmonic studies to predict the transmission or reflection spectrum.

## Device fabrication
The ten experimental samples are fabricated by additive manufacturing (3D-printing) using selective laser sintering with a manufacturing precision of ±0.2 mm. The material of choice is polyamide (PA 2200 from EOS). We manufacture four samples with high-transmission glass side walls to facilitate experiments using refracto-vibrometry. The glass is bonded to the sample and the sample is hermetically sealed.

## Measurement
The complex transmission (and reflection) coefficients of the samples are measured using an AED 1200−AcoustiTube transmission tube with a diameter of 40 mm. The transmission coefficient and the transmission loss are calculated by applying the two-load method with four microphones according to the transfer matrix method. The absorption coefficients of the symmetry-reduced cavities are measured using an AED 1000−AcoustiTube impedance tube with a diameter of 40 mm.

## Visualization
We use refracto-vibrometry to visualize the sound pressure field inside the cavity. A laser Doppler scanning vibrometer PSV-500 from Polytec is used to measure the changes in the refraction index of the fluid, which is proportional to the acoustic pressure variation within the cavity. Overall, 225 measurement points were sequentially recorded with a sampling frequency of 50 kHz for a duration of 2 ms, while the measurements were triggered by the sinusoidal sound generator. To ensure an optimal signal-to-noise ratio, a highly reflective sheet was placed against the rigid surface behind the sample to improve diffuse light reflection. Note that LDV is usually used for surface normal vibration measurements of structures but captures the pressure wave-induced variation in the refraction index when measured against a rigid surface. In the case of a low-vibration surface, the velocity measurement from the LDV is dominated by the dynamic phase caused by the sound pressure fluctuations and the changed refractive index of the acoustic medium along the traveling path of the light. To ensure the required rigidity, a single point LDV (Polytec PDV-100) measured the surface vibration of the rigid surface from the opposite direction. The surface velocities were found to be orders of magnitude smaller than the signal of the scanning PSV500, confirming that the acoustic pressure dominates the measured results. As the LDV works up to frequencies of 1 MHz, the frequency range is not a limiting factor for pressure field mapping. The size of the structure can be much smaller than those presented in this article and is only limited by the focal point of the laser (25 μm). We use such large structures here because we need to measure the transmission/reflection spectra using an impedance tube with a diameter of 40 mm. Since the Helmholtz equation is linearly scalable, our results can be transferred to different frequency ranges.

## Data availability
The data used in this study are available in the figshare database under [https://doi.org/10.6084/m9.figshare.24211515].

## Code availability
The codes used in this study are available in the figshare database under [https://doi.org/10.6084/m9.figshare.24211515]. Additionally, we host a COMSOL server, where we provide free access to vibro-

acoustic applications: https://apps.vib.ed.tum.de:2037/app-lib. We particularly created an application to give people an understanding of BICs and their influence on sound attenuation: https://apps.vib.ed.tum.de:2037/app/BIC_TL_EF_App_V02_mph?id=0012.

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

## Author contributions

F.K., S.M., S.O., and D.A.P. conceived the project. F.K., Y.K.C., L.H., and D.A.P. designed the geometry and modeled the physics behind it. F.K. and M.M. made samples and performed measurements to obtain the transmission spectra and the visualization of the BIC. D.A.P. and Y.K.C. advised the modeling and experimental process. J.D.S. implemented and uploaded the BIC APP. All authors discussed the results. F.K. prepared the paper with contributions from all authors. M.M., S.M., S.O., and D.A.P. edited the paper.

## Funding

## Competing interests

The authors declare no competing interests.
