## [Peer Review File · Nature Communications]

Reviewers' Comments:

Reviewer #1:

Remarks to the Author:

The authors theoretically investigated and experimentally demonstrated the existence of FW-BICs by measuring the transmission spectrum. By scanning the size ratio of rectangular resonators, BIC is converted into QBIC, which is usually manifested by the Fano resonance in the transmission spectrum. Besides the transmission spectrum, the authors image the pressure field distribution of QBICs based on FW-BICs by using laser Doppler vibrometry, proving the direct evidence of the existence of FW-BICs. Although this work presents some interesting results, some concerns must be addressed

(1) The biggest selling point lies in directly imaging the QBICs based on FW-BIC. In Fig.4, indeed, the measured pressure distribution of QBICs matches very well with the simulated one. Due to the thermal-viscous loss, the pressure field enhancement is not as high as the simulated one if the lossless system is considered. Therefore, to further push the applications of QBICs, real intensities instead of normalized intensities of the pressure field of QBICs shall be presented because the maximum pressure field plays a central role in governing acoustic-matter interactions, such as acoustic emissions. Therefore, I may suggest authors provide more data with the real intensities of the pressure field at different size ratios to demonstrate that the pressure field enhancement reaches the maximum when the size ratio is close to the critical value forming FW-BICs. With real information on pressure intensity, it may provide useful guidance for designing acoustic sources.

(2) In Fig.2c-d, a good agreement can be found between measurement and simulation. It is well known that in real systems, there are thermo-viscous losses that may influence the transmission significantly. Do authors consider the thermo-viscous losses in the simulation to match the experiments? If yes, they need to provide more details on how thermo-viscous losses affect the simulated transmission.

(3) Since the authors show the pressure imaging of FW-BICs in Fig.4, I may suggest the authors also do the same thing on the BICs formed in structures of Fig.5d. By doing this, the authors may know which structure gives the best pressure confinement of QBICs.

(4) In Fig.2c, Fig.4e-f, Fig.4h-i, what does it mean for y-axis, $T[-]$, $R[-]$, $\alpha[-]$? Usually, the unit is included in the bracket.

Reviewer #2:

Remarks to the Author:

The authors demonstrate theoretically and experimentally the existence of Friedrich-Wintgen (FW) type bound in continuum states (BICs) in an open acoustic cavity. The BIC is revealed in the transmission coefficient under the shape of a Fano resonance (quasi-BIC) whose linewidth vanishes when approaching the geometrical parameters of the cavity satisfying the BIC condition. They show the best conditions to observe the QBIC in presence of thermoviscous losses. Moreover, by using a laser Doppler vibrometry (LDV) technique, they are able to reveal for the first time the map of the pressure field associated with the BIC inside the cavity. Finally, they present the design of new type high-quality BIC relying on reduced symmetry and avoiding the excitation of unwanted resonances in the cavity. The agreement between theory and experiment is very accurate throughout the manuscript.

The paper is generally well presented and contains some interesting and new results. It is timely because of the continuing interest in BICs and their applications based on the high sensitivity of the resulting Fano resonances to the environment. However, the cavity designed in this paper and the associated BICs and their analysis have some similarities with those presented in recent papers. The main novelty of this manuscript is the observation of the BIC pressure field inside the cavity by using LDV. This is an interesting proposal that deserves to be published. But from the point of view of physical originality or attraction for a broad audience, the paper would probably not reach the level required for acceptance in Nature Communications. Perhaps, another of the Nature journals would be more appropriate.

A few minor remarks are mentioned for the attention of the authors.

- A short explanation about FW BIC can be useful, as well as on the physical principle of LDV.
- In page 5, the sentence "a reduction of a factor of $k = 3.41$ between the inlet and the outlet sound pressure amplitudes was noticed" needs a better explanation.
- In Figure S3, one cannot clearly see the evolution of L_x on the curves, so some precision would be helpful.
- In section S5, Figures S6 are referred to as Fig. 6 (Fig. 6(a), 6(b) and 6(c)). In Fig. S6, two panels are labeled b instead of b and c. Please correct

Reviewer #3:

Remarks to the Author:

The authors demonstrate the existence of a Friedrich-Wintgen BIC in an open acoustic cavity by numerical simulations and experiments, and indirectly measure the pressure field of the transparent open cavity hosting a Friedrich-Wintgen BIC using laser Doppler vibrometry. The specific questions are as follows:

- (1) The models in this article are very similar to previous works (such as Phys. Rev. Applied 18, 054021), the innovative of the work should be further explained, and there is no detailed theory analysis in this article.
- (2) As far as I know, it is not the first time to indirectly measure the acoustic pressure in air by the laser Doppler vibrometry. On the other hand, for the model size in this article, it is possible to directly measure the pressure field by such as 1/4" microphone.
- (3) Could authors explain further the black dotted line in the Fig.1(b)(c)? How do authors calculate them through the coupled mode theory?
- (4) Does the technique involved in this article using laser Doppler vibrometry require the structural size of the sample? When it comes to practicality, does this technique work on smaller structures?

Response to the Reviewers

Journal:

Manuscript No.:

NCOMMS-22-48526A-Z

Title:

Realistic prediction and engineering of high-Q modes to implement stable Fano resonances in acoustic devices

Authors:

Felix Kronowetter, Marcus Maeder, Yan Kei Chiang, Lujun Huang, Johannes Schmid, Sebastian Oberst, David A. Powell, and Steffen Marburg

General Remarks

Dear Reviewers:

We would like to express our appreciation for the efforts made by the reviewers to improve our paper. The comments are very constructive and helpful in improving the quality and presentation of the paper. We have done a substantial amount of work to incorporate the suggestions and address the concerns raised by the reviewers. In response to reviewer #1's request, we have performed extensive numerical simulations and put considerable effort into fabricating new samples and performing additional experiments. In response to the requests of reviewers #2 and #3, we have added more data and more detailed explanations to the manuscript to highlight the novelty, innovation, and scientific impact of our work. We have added a complete theoretical model in response to reviewer #3. We have incorporated the suggestions and addressed the concerns raised by the reviewers. **Red-colored text indicates changes made to the manuscript**, while **blue-colored text highlights the authors' responses**.

Specific Response to Reviewer #1:

The authors theoretically investigated and experimentally demonstrated the existence of FW-BICs by measuring the transmission spectrum. By scanning the size ratio of rectangular resonators, BIC is converted into QBIC, which is usually manifested by the Fano resonance in the transmission spectrum. Besides the transmission spectrum, the authors image the pressure field distribution of QBICs based on FW-BICs by using laser Doppler vibrometry, proving the direct evidence of the existence of FW-BICs. Although this work presents some interesting results, some concerns must be addressed.

Thank you for your feedback and helpful comments. We have tried to incorporate all of them to a satisfactory level.

1. The biggest selling point lies in directly imaging the QBICs based on FW-BIC. In Fig. 4, indeed, the measured pressure distribution of QBICs matches very well with the simulated one. Due to the thermal-viscous loss, the pressure field enhancement is not as high as the simulated one if the lossless system is considered. Therefore, to further push the applications of QBICs, real intensities instead of normalized intensities of the pressure field of QBICs shall be presented because the maximum pressure field plays a central role in governing acoustic-matter interactions, such as acoustic emissions. Therefore, I may suggest authors provide more data with the real intensities of the pressure field at different size ratios to demonstrate that the pressure field enhancement reaches the maximum when the size ratio is close to the critical value forming FW-BICs. With real information on pressure intensity, it may provide useful guidance for designing acoustic sources.

We have changed Fig. 4 and the corresponding text to show the actual pressure values. We use absolute pressure from here on to be independent of the phase (section: Results, Fig. 4, pages: 6 and 7):

The visualized sound pressure field of the QBIC within the experimental set-up is presented in Fig. 4a. In addition, we show a mapping of the absolute values of the pure measurement data in Fig. 4b for better comparison with the numerical data displayed in Fig. 4e. We observe that the measurement is in good agreement with the numerical prediction. As expected, the pressure maxima are located at both the edges and the center of the cavity. Minor inaccuracies within the numerical prediction are due to real losses and uncertainties within the experiments, i.e. imprecise alignment of the parts or imperfect plane wave excitation. We note only a slight pressure fluctuation in the center, just below the marked node in Fig. 4a. The position of the LDV for this measurement point is normal incidence to the glass. This results in a lower signal-to-noise ratio, which leads to higher measurement errors. We obtain a stable image of the QBIC that is in good agreement with the simulated sound pressure field. Thus, we present the visual evidence of a QBIC.

In addition, we have added plots of the maximum pressure inside the cavity versus cavity length and frequency for all configurations presented in the manuscript to see where the real maximum field enhancement occurs. We have also measured the cavity with $L_x = 170$ mm excited at several frequencies to observe the evolution of the modal field and to determine the frequency with the highest pressure enhancement. We have also provided additional data with different size ratios of the fully reduced cavity, shown in Fig. 5c, to demonstrate the parameter configuration where the pressure field enhancement reaches its maximum. We have chosen this configuration to experimentally determine where the maximum pressure enhancement occurs. The results for the configuration with the highest pressure enhancement are presented in the manuscript as they provide seminal insights for the design of QBICs (section: Results, Pressure Enhancement, Fig. 5, pages: 6 - 8):

Fig. 4. Sound pressure field visualization. **a** Visualization of the real part of the sound pressure field inside the cavity. The red color indicates a pressure maximum and the blue color a minimum, respectively. **b** Absolute value of the measured pressure field. **c** Superposition of the four most dominant modes (modes with the highest FFT coefficients). **d** Superposition of the next four modes with high coefficients. **e** Simulation results. The colored scales display the absolute values of the pressure in Pa, with the pressure being normalized to the incident pressure field.

Figs. 5a to 5c show three schematics of the previously studied resonant cavities reduced in size by applying the principle of mirror sources. We cut the geometry in half at a particular axis of symmetry (marked by the red lines in the schematics mentioned above). The axis of symmetry has to be chosen with care since the BIC mode needs to be sustained and all anti-symmetric modes with respect to this axis need to be suppressed. Fig. 5c depicts the schematic of the fully reduced geometry. We manufacture three additional samples for impedance tube measurements, shown in Fig. 5d. Fig. 5e shows the numerically obtained absorption coefficients of the configurations in Figs. 5a to 5c with $L_x = 85$ mm and also of the BIC configuration in Fig. 5c determined at $L_x = 77$ mm. The more we reduce the geometry, the more unwanted modes are removed. Accordingly, we observe a clear peak in the absorption spectrum. We perform impedance tube measurements to validate our observations. The measured absorption coefficients are presented in Fig. 5f for the fully reduced configuration, see Fig. 5c, and in Fig. 5g for the configuration shown in Fig. 5b. The measurement results agree with our prediction. We observe the collapse of the Fano resonance (yellow and dashed purple lines in Fig 5f), the sharp peak of the Fano resonance with no further absorption in the spectrum shown (blue and dashed red lines), and thus prove the existence of a Friedrich-Wintgen BIC based on the concept of fully reduced symmetry without exciting unwanted modes. Fig. 5h shows the unexcited QBIC mode for a rectangular cavity with $L_x = 170$ mm. The sound pressure field of the fully reduced geometry framed by the black lines is depicted in Fig. 5i. For comparison, we extrapolate the reduced sound pressure field onto the rectangular cavity. We infer from comparing Fig. 5h

Fig. 5. Experimental verification of symmetry-reduced BICs. **a** Schematic of a resonant cavity similar to that shown in Fig. S 1. The z -axis is perpendicular to the xy -plane. The length L_y is set to 160 mm and the diameter d of the pipe to 40 mm. We halve the cavity at the axis of symmetry (highlighted in red) based on the principle of mirror sources and thus obtain a cavity length of $L_x = 85$ mm, which corresponds to half the length of the configuration with $L_x = 170$ mm. **b** Further reduction of the configuration in **a** using the principle of mirror sources once again. **c** Fully reduced configuration. **d** Manufactured samples for configurations **b** ($L_x = 85$ mm) and **c** ($L_x = 85$ mm for the top sample and $L_x = 77$ mm for the bottom sample). **e** Absorption coefficient α in the frequency range 1900–2250 Hz. The blue line represents the result of the numerical simulation of the BIC configuration of **c** with $L_x = 77$ mm. The orange dotted line, the yellow dash-dotted line and the purple dashed line represent the numerical results of $L_x = 85$ mm for the configurations **a**, **b**, and **c**, respectively. **f** Absorption coefficient α in the frequency range 1900–2250 Hz. The blue and the yellow lines represent the results of the numerical simulation of the BIC configuration of **c** with $L_x = 85$ mm and $L_x = 77$ mm, respectively. The orange and the purple dashed lines represent the measurement results of configuration **c** with $L_x = 85$ mm and $L_x = 77$ mm. **g** The computed and measured absorption coefficients for configuration **b** are represented by the blue and orange dashed lines, respectively. **h** Sound pressure field of the BIC mode inside the cavity with $L_x = 170$ mm. **i** Sound pressure field of the BIC mode inside the fully reduced cavity with $L_x = 85$ mm for configuration **c**, framed by the black lines and extrapolated to the rectangular cavity. **j** Measured absolute sound pressure inside the fully reduced cavity with $L_x = 67$ mm excited at 2315 Hz. The color scale represents the absolute pressure in Pa.

to Fig. 5i that identical sound pressure distributions are obtained. Finally, we manage to preserve the BIC mode although the geometry is reduced significantly and thus present a new type of fully reduced high-Q mode without exciting unwanted modes.

Pressure enhancement. Impedance tube studies allow us to extract values for reflection spectra and absorption, but not for cavity pressure enhancement. The missing cavity pressure information is provided by the LDV. The numerical and experimental procedure to determine the maximum pressure is presented in detail in Supplementary Information Section 6. Fig. 5j shows the measured absolute pressure field inside the fully reduced cavity with $L_x = 67$ mm excited at 2315 Hz. This configuration leads to the maximum pressure enhancement possible for this structure with a measured peak value of 36.74 Pa. We additionally measure the sound pressure fields of the fully reduced cavity with $L_x = 64$ mm and $L_x = 70$ mm to prove the existence of a pressure peak. The measurements show peak pressures of 26.08 Pa and 32.26 Pa, respectively. To demonstrate the magnitude of the pressure enhancement, we measure the pressure field inside the full cavity with $L_x = 170$ mm for several frequencies. The maximum pressure is 12.61 Pa. Thus, the fully reduced cavity leads to the highest pressure enhancement of the investigated Friedrich-Wintgen BIC by a factor of about three compared to the full cavity.

Finally, we extract the sound pressure field of a QBIC mode. The sound pressure field inside the cavity gives us accurate information about the influence of losses and hence the stability and confinement of the mode. We also show that LDV measurements are a powerful tool for predicting the maximum pressure enhancement of high-Q modes, can resolve even small deviations from numerical predictions, and thus provide seminal guidance for the application of QBICs. Thus, we present the realistic sound pressure field of a stable high-Q mode, enabling further analysis and its application to acoustic devices.

All other results are shown in Supplementary Information Section S 6:

Pressure enhancement

The configuration with the highest pressure field enhancement is determined by parameter studies. Therefore, we vary the cavity lengths of the configurations depicted in Fig. 1a, Fig. 5a and Fig. 5b. The corresponding plots are shown in Fig. S 11.

Fig. S 11. Simulated pressure mappings. **a** Maximum absolute sound pressure inside the rectangular cavity shown in Fig. 1a. The cavity length is varied from $L_x = 145 - 185$ mm in 1 mm and 1 Hz steps in the frequency range 1900 - 2250 Hz. **b** Maximum absolute sound pressure inside the reduced cavity presented in Fig. 5a with cavity length $L_x = 67.5 - 92.5$ mm in the same frequency range. **c** Maximum absolute sound pressure inside the further reduced cavity (Fig. 5b). The white crosses indicate the maxima.

Pressure enhancement can be observed in certain regions around the BIC configurations. We can also see that the maximum pressure enhancement does not occur directly adjacent to the BIC as it would without losses, i.e., see Fig. S 8. Thermo-viscous losses shift the maximum enhancement away from the BIC configuration. The pressure peaks are 5.50 Pa, 9.54 Pa, and 12.89 Pa. It can be said that the more the geometry is reduced and thus the antisymmetric modes are suppressed, the higher the pressure enhancement. The visualized pressure fields for the full cavity are shown in Fig. S 12.

Fig. S 12. Visualized pressure fields of the full cavity. a - f Absolute sound pressure inside the rectangular cavity shown in Fig. 1a. with $L_x = 170$ mm excited at 2070 Hz, 2075 Hz, 2077 Hz, 2079 Hz, 2081 Hz, and 2094 Hz, respectively. All color scales represent the absolute pressure in Pa.

The corresponding maximum pressure values in Figs. S 12a to 12f are 12.04 Pa, 12.61 Pa, 12.44 Pa, 12.02 Pa, 11.32 Pa, and 8.59 Pa, respectively. Therefore, we can say that the maximum pressure enhancement occurs at 2075 Hz. The evolution of the modal field is also demonstrated. The dark blue line

representing the pressure nodes extends in the y -direction, the more the configuration deviates from the BIC configuration.

We numerically determine the configuration with the highest pressure gain by plotting the maximum absolute sound pressure inside the cavity as the cavity length is varied in the 1900 - 2400 Hz frequency range and excited by a plane wave of 1 Pa. This is shown in Fig. S 13 for the fully reduced cavity presented in Fig. 5c.

Fig. S 13. Simulated pressure mappings. **a,b** Maximum absolute sound pressure and absorption inside the cavity shown in Fig. 5c. The cavity length is varied from $L_x = 67.5 - 92.5$ mm in 1 mm and 1 Hz steps in the frequency range 1900 - 2400 Hz. **c,d** Maximum absolute sound pressure and absorption at finer resolution. The cavity length is varied from $L_x = 66 - 68$ mm in 0.1 mm and 0.1 Hz steps in the frequency range 2290 - 2330 Hz. The white crosses indicate the maxima.

The BIC is visible as the dark blue dot in the center of the red lines in Figs. S 13a to 13b. Pressure enhancement can be observed in certain regions around the BIC configurations. The smaller the cavity length, the higher the frequency of the enhancement and vice versa. It can be seen that the maximum pressure enhancement and absorption occurs at a cavity length of $L_x = 67$ mm. We fabricate three additional samples of the fully reduced cavity ($L_x = 64, 67,$ and 70 mm) to experimentally validate our numerical predictions, see Fig. S 14.

Fig. S 14. LDV samples. Printed samples of the fully reduced cavity with high-transmission glass mounted as side panels.

We then measure the sound pressure field inside the cavity for several frequencies to determine the configuration (frequency and cavity length) where the maximum absolute sound pressure is found. The visualized pressure fields for the fully reduced cavity with $L_x = 64$ mm, $L_x = 67$ mm, and $L_x = 70$ mm are shown in Fig. S 15.

Fig. S 15. Visualized pressure fields of the fully reduced cavity. **a - c** Absolute sound pressure inside the fully reduced cavity shown in Fig. 5c with $L_x = 64$ mm excited at 2335 Hz, 2345 Hz, and 2353 Hz. **g - i** Absolute sound pressure inside the fully reduced cavity with $L_x = 70$ mm excited at 2267 Hz, 2272 Hz, and 2277 Hz. All color scales represent the absolute pressure in Pa.

The data are not normalized to the incident pressure field, but the actual pressure values are shown. The corresponding maximum pressure values in Figs. S 15a to 15i are 25.61 Pa, 26.07 Pa, 24.80 Pa, 36.11 Pa, 36.74 Pa, 33.98 Pa, 31.83 Pa, 32.26 Pa, and 28.12 Pa, respectively. Therefore, we can say that the maximum pressure enhancement for the cavity with $L_x = 64$ mm occurs at 2345 Hz and for the cavity with $L_x = 70$ mm at 2272 Hz. The maximum pressure enhancement occurs at $L_x = 67$ mm at 2315 Hz and reaches 36.74 Pa. This is similar to the results shown in Fig. S 13c.

We compare this maximum pressure value to the sound pressure fields of the fully reduced cavity with $L_x = 64$ mm and $L_x = 70$ mm to prove the existence of a pressure peak. The measurements show peak pressures of 26.07 and 32.26 Pa, respectively. To demonstrate the magnitude of the pressure enhancement, we also compare it to the pressure field inside the full cavity with $L_x = 170$ mm for several frequencies.

The maximum pressure of the full cavity is 12.61 Pa. Thus, the fully reduced cavity leads to the highest pressure enhancement of the investigated Friedrich-Wintgen BIC by a factor of about three.

- In Fig.2c-d, a good agreement can be found between measurement and simulation. It is well known that in real systems, there are thermo-viscous losses that may influence the transmission significantly. Do authors consider the thermo-viscous losses in the simulation to match the experiments? If yes, they need to provide more details on how thermo-viscous losses affect the simulated transmission.

We have included thermo-viscous losses in our simulations. To address the effect of losses on the transmission and reflection spectra, we have added a more detailed explanation in section S 4 in the Supplementary Information (section: S 4, pages: 7 and 8):

The effect of thermo-viscous losses on the Fano peaks in the transmission spectra is shown in Fig. S 7.

Fig. S 7. Transmission spectra with and without losses. **a** Transmission coefficient in the 1900 - 2250 Hz frequency range. The solid blue and dashed yellow lines represent the results of $L_x = 165$ mm with and without thermo-viscous losses, respectively. The results of $L_x = 170$ mm are shown by the red and purple lines. **b** Transmission loss of $L_x = 165$ mm and $L_x = 170$ mm with and without losses. The coloring of the lines is identical to Fig. S 7a.

Thermo-viscous losses significantly reduce the transmission coefficient due to increased absorption. Nevertheless, the transmission goes to zero at the frequency of the QBIC, see Fig. S 7a. The maxima of the Fano peaks in the TL also decrease including the losses. We observe a reduction from ≈ 42 dB to ≈ 17 dB ($L_x = 165$ mm) and from ≈ 43 dB to ≈ 24 dB ($L_x = 170$ mm). The losses have a more significant effect on the amplitude of the Fano peak of the TL the closer we are to the BIC configuration. In addition, due to thermo-viscous losses, the Fano peaks are shifted to lower frequencies by about 2 Hz. To illustrate the effect of losses on pressure field enhancement, the maximum absolute sound pressure is plotted against cavity length and frequency in Fig. S 8.

Fig. S 8. Pressure mapping (without losses). Maximum absolute sound pressure inside the rectangular cavity shown in Fig. 1a. The cavity length is varied from $L_x = 157 - 163$ mm in 0.1 Hz steps in the frequency range 2135 - 2160 Hz.

We observe amplified sound pressure up to 160 dB when excited at 1 Pa and narrow Fano peaks near the BIC. Figs. S 6 to S 8 illustrate the importance of considering thermo-viscous losses in our simulations.

3. Since the authors show the pressure imaging of FW-BICs in Fig.4, I may suggest the authors also do the same thing on the BICs formed in structures of Fig.5d. By doing this, the authors may know which structure gives the best pressure confinement of QBICs.

We have manufactured three additional configurations of the fully reduced cavity, shown in Fig. 5c, and have visualized the pressure fields. We have chosen this configuration to experimentally determine where the maximum pressure enhancement occurs. The results of the configuration with the highest pressure enhancement are presented in the manuscript as they provide seminal insights for the design of QBICs, while the results for the other configurations appear in Supporting Information S 6. See the answer to your first point for more details.

4. In Fig.2c, Fig.4e-f, Fig.4h-i, what does it mean for y-axis, T[-], R[-], alpha[-] ? Usually, the unit is included in the bracket.

We have used [-] to indicate, that the corresponding quantities are dimensionless, but have removed them for a better understanding.

Specific Response to Reviewer #2:

The authors demonstrate theoretically and experimentally the existence of Friedrich-Wintgen (FW) type bound in continuum states (BICs) in an open acoustic cavity. The BIC is revealed in the transmission coefficient under the shape of a Fano resonance (quasi-BIC) whose linewidth vanishes when approaching the geometrical parameters of the cavity satisfying the BIC condition. They show the best conditions to observe the QBIC in presence of thermoviscous losses. Moreover, by using a laser Doppler vibrometry (LDV) technique, they are able to reveal for the first time the map of the pressure field associated with the BIC inside the cavity. Finally, they present the design of new type high-quality BIC relying on reduced symmetry and avoiding the excitation of unwanted resonances in the cavity. The agreement between theory and experiment is very accurate throughout the manuscript.

The paper is generally well presented and contains some interesting and new results. It is timely because of the continuing interest in BICs and their applications based on the high sensitivity of the resulting Fano resonances to the environment. However, the cavity designed in this paper and the associated BICs and their analysis have some similarities with those presented in recent papers. The main novelty of this manuscript is the observation of the BIC pressure field inside the cavity by using LDV. This is an interesting proposal that deserves to be published. But from the point of view of physical originality or attraction for a broad audience, the paper would probably not reach the level required for acceptance in Nature Communications. Perhaps, another of the Nature journals would be more appropriate.

To emphasize the novelty, innovation, and scientific impact of our work, we have modified the abstract (section: Abstract, page: 1):

Bound states in the continuum (BICs) coupling into the propagating spectrum lead to the formation of quasi-BICs (QBICs) with high quality (Q) factor. These high-Q modes typically manifest as Fano resonances in the transmission or reflection spectrum. Fano resonances are highly sensitive, and even a small change in geometric or environmental variables results in a significant change in the transmission spectrum. This presents a challenge in predicting stable Fano peaks for realistic applications. In addition, we do not yet have any information on whether or not a large field enhancement occurs when thermo-viscous losses are taken into account in real acoustic devices. In this work, we demonstrate the existence of a Friedrich-Wintgen BIC in an open acoustic cavity by theory and experiment. In agreement with theoretical predictions, the appearance of the BIC is characterized by the vanishing line width of the Fano resonance in the measured transmission spectrum. We map the pressure field of the transparent open cavity hosting a Friedrich-Wintgen BIC using laser Doppler vibrometry, which is the first reported visualization of an acoustic BIC. Mapping the pressure field of the BIC is a new technique for extracting real sound pressure values of high-Q modes. From the pressure values, we accurately infer the confinement of the BIC and thus its transmission or reflection properties. Furthermore, based on our results, we design a new type of symmetry-reduced BIC and analyze its absorption spectrum. Laser Doppler vibrometer (LDV) measurements provide the missing field enhancement data. We achieve a field enhancement by a factor of about three compared to the original cavity. The main findings are that LDV measurements are a powerful tool for predicting the maximum pressure enhancement of high-Q modes, and thus a new technique for obtaining the missing field enhancement data. The presented results facilitate the future applications of BICs in acoustics as high-intensity sound sources, filters, and sensors.

We have added a more detailed explanation of the innovation and of the importance of our findings (section: Introduction, page: 2):

We further present the first direct visualization of a Friedrich-Wintgen BIC using laser Doppler vibrometry as a pressure field mapping technique. We use laser Doppler vibrometry measurements to obtain a complete mapping of the sound pressure field to better understand BIC formation in the presence of realistic losses. The reflection and transmission spectra are obtained using microphones, but do not provide information about the exact pressure enhancement inside the cavity. The pressure distribution inside the cavity is needed to develop high-performance acoustic devices based on BICs, such as acoustic sources and sound lasers. Because BICs are extremely sensitive, any perturbation of the high-Q mode, energy extraction, or backscattering from microphones will degrade the BIC, so we use this technique to avoid any perturbation of the pressure field. By mapping the sound pressure field of the BIC, we have direct access to the actual pressure values and thus to the critical information of when and where the pressure magnitude reaches a maximum. The interaction between the localized thermo-viscous losses and the concentrated high intensity fields of the BIC is the key to deter-

mining the achievable Q-factor and pressure field enhancement. We facilitate the accurate analysis of high-Q modes in the presence of realistic losses, determine the configuration with maximum pressure enhancement, and enable the future application of high-Q Fano resonances to acoustic devices.

A few minor remarks are mentioned for the attention of the authors.

1. A short explanation about FW BIC can be useful, as well as on the physical principle of LDV.

We have added an explanation of the formation mechanism (section: S 1, pages: 1 - 3). For a detailed response, please see Specific Response to Reviewer #3, 1.). We have also added a short description of the analytical solution to predict the eigenfield profile of the Friedrich-Wintgen BIC (section: Methods, page: 9):

Analytical Model. The eigenfield profile of the Friedrich-Wintgen BIC can be predicted from the following equation

$$\psi_{BIC}(x, y) \approx \cos \theta \psi_{31}(x, y) + \sin \theta \psi_{13}(x, y)$$

where

$$\cos \theta = \frac{A}{\sqrt{A^2 + B^2}} \quad , \quad \sin \theta = \frac{B}{\sqrt{A^2 + B^2}},$$

$$A = -\sqrt{\frac{L_y}{2\pi^2 L_x}} \left[\sin\left(\frac{\pi(Ly+1)}{L_y}\right) - \sin\left(\frac{\pi(Ly-1)}{L_y}\right) \right],$$

$$B = \sqrt{\frac{2}{L_x L_y}}.$$

See Supplementary Information Section 1 for a complete theoretical analysis.

We have added an explanation of the physical principle of the LDV (section: Results, Visualization of QBIC, page: 4):

To conduct the refracto-vibrometry, a laser Doppler scanning vibrometer PSV-500 from Polytec¹ is used to measure the changes of the refractive index of the fluid, which is proportional to the acoustic pressure variation within the cavity [43-46]:

$$v(\omega) = \omega \frac{1}{\gamma p_0} \frac{n_0 - 1}{n_0} \int_L p(l, \omega) dl \quad (1)$$

where ω is the angular frequency, v is the LDV velocity, p is the sound pressure, n_0 is the refractive index of air at standard atmospheric pressure, p_0 is the static atmospheric pressure, and γ is the specific heat capacity ratio of air. The basic principle of the LDV is based on the well-known Doppler shift. The pressure waves inside the cavity slightly shift the phase of the emitted monochromatic laser light. The superimposition of the reflected and emitted laser light produces a speckle pattern on the photodetector, which allows the measurement of the corresponding frequency of the pressure waves and the change in the refractive index. The latter is proportional to the sound pressure inside the cavity. This makes it possible to visualize the corresponding pressure distribution.

We have also added a more detailed explanation of the visualization and its limits (section: Methods, pages: 9 and 10):

Visualization. We use refracto-vibrometry to visualize the sound pressure field inside the cavity. A laser Doppler scanning vibrometer PSV-500 from Polytec is used to measure the changes in the refraction index of the fluid, which is proportional to the acoustic pressure variation within the cavity. Overall, 225 measurement points were sequentially recorded with a sampling frequency of 50 kHz for a duration

¹Polytec Gmbh, Waldbronn, Germany

of 2 ms, while the measurements were triggered by the sinusoidal sound generator. To ensure an optimal signal-to-noise ratio, a highly reflective sheet was placed against the rigid surface behind the sample to improve diffuse light reflection. Note that usually a LDV is used for surface normal vibration measurements of structures but captures the pressure wave-induced variation in the refraction index when measured against a rigid surface. In the case of a low-vibration surface, the velocity measurement from the LDV is dominated by the dynamic phase, caused by the sound pressure fluctuations, and the changed refractive index of the acoustic medium, along the traveling path of the light. To ensure the required rigidity, a single point LDV (Polytec PDV-100) measured the surface vibration of the rigid surface from the opposite direction. The surface velocities were found to be orders of magnitude smaller than the signal of the scanning PSV500, confirming that the acoustic pressure dominates the measured results. As the LDV works up to frequencies of 1 MHz, the frequency range is not a limiting factor for pressure field mapping. The size of the structure can be much smaller than those presented in this article and is only limited by the focal point of the laser ($25\ \mu\text{m}$). We use such large structures here because we need to measure the transmission/reflection spectra using an impedance tube with a diameter of 40 mm. Since the Helmholtz equation is linearly scalable, our results can be transferred to different frequency ranges.

2. In page 5, the sentence “a reduction of a factor of $k = 3.41$ between the inlet and the outlet sound pressure amplitudes was noticed” needs a better explanation.

We have removed that part and have introduced new results regarding the pressure amplification inside the cavity. The new results also show the importance of our findings. Please see Specific Response to Reviewer #1, 1.).

3. In Figure S3, one cannot clearly see the evolution of L_x on the curves, so some precision would be helpful.

We have added an explanation of the evolution of L_x (section: S 3, Fig. S 5, page: 6):

Fig. S 5. Complex eigenfrequencies for varying L_x . **a** The spectrum of propagating waves dominated by the pipe is marked by the region colored blue. Cavity resonances are highlighted by the region colored red. BIC 1 and BIC 2 are marked by the red crosses. **b** Evolution over L_x of the interacting modes that form BIC 1 & 2.

4. In section S5, Figures S6 are referred to as Fig. 6 (Fig. 6(a), 6(b) and 6(c)). In Fig. S6, two panels are labeled b instead of b and c. Please correct

We have changed the labels.

Specific Response to Reviewer #3:

The authors demonstrate the existence of a Friedrich-Wintgen BIC in an open acoustic cavity by numerical simulations and experiments, and indirectly measure the pressure field of the transparent open cavity hosting a Friedrich-Wintgen BIC using laser Doppler vibrometry. The specific questions are as follows:

1. The models in this article are very similar to previous works (such as Phys. Rev. Applied 18, 054021), the innovative of the work should be further explained, and there is no detailed theory analysis in this article.

To emphasize the novelty, innovation, and scientific impact of our work, we have modified the abstract (section: Abstract, page: 1):

Bound states in the continuum (BICs) coupling into the propagating spectrum lead to the formation of quasi-BICs (QBICs) with high quality (Q) factor. These high-Q modes typically manifest as Fano resonances in the transmission or reflection spectrum. Fano resonances are highly sensitive, and even a small change in geometric or environmental variables results in a significant change in the transmission spectrum. This presents a challenge in predicting stable Fano peaks for realistic applications. In addition, we do not yet have any information on whether or not a large field enhancement occurs when thermo-viscous losses are taken into account in real acoustic devices. In this work, we demonstrate the existence of a Friedrich-Wintgen BIC in an open acoustic cavity by theory and experiment. In agreement with theoretical predictions, the appearance of the BIC is characterized by the vanishing line width of the Fano resonance in the measured transmission spectrum. We map the pressure field of the transparent open cavity hosting a Friedrich-Wintgen BIC using laser Doppler vibrometry, which is the first reported visualization of an acoustic BIC. Mapping the pressure field of the BIC is a new technique for extracting real sound pressure values of high-Q modes. From the pressure values, we accurately infer the confinement of the BIC and thus its transmission or reflection properties. Furthermore, based on our results, we design a new type of symmetry-reduced BIC and analyze its absorption spectrum. Laser Doppler vibrometer (LDV) measurements provide the missing field enhancement data. We achieve a field enhancement by a factor of about three compared to the original cavity. The main findings are that LDV measurements are a powerful tool for predicting the maximum pressure enhancement of high-Q modes, and thus a new technique for obtaining the missing field enhancement data. The presented results facilitate the future applications of BICs in acoustics as high-intensity sound sources, filters, and sensors.

We have added a more detailed explanation of the innovation and of the importance of our findings (section: Introduction, page: 2):

We further present the first direct visualization of a Friedrich-Wintgen BIC using laser Doppler vibrometry as a pressure field mapping technique. We use laser Doppler vibrometry measurements to obtain a complete mapping of the sound pressure field to better understand BIC formation in the presence of realistic losses. The reflection and transmission spectra are obtained using microphones, but do not provide information about the exact pressure enhancement inside the cavity. The pressure distribution inside the cavity is needed to develop high-performance acoustic devices based on BICs, such as acoustic sources and sound lasers. Because BICs are extremely sensitive, any perturbation of the high-Q mode, energy extraction, or backscattering from microphones will degrade the BIC, so we use this technique to avoid any perturbation of the pressure field. By mapping the sound pressure field of the BIC, we have direct access to the actual pressure values and thus to the critical information of when and where the pressure magnitude reaches a maximum. The interaction between the localized thermo-viscous losses and the concentrated high intensity fields of the BIC is the key to determining the achievable Q-factor and pressure field enhancement. We facilitate the accurate analysis of high-Q modes in the presence of realistic losses, determine the configuration with maximum pressure enhancement, and enable the future application of high-Q Fano resonances to acoustic devices.

We have added an explanation of the formation mechanism (section: S 1, pages: 1 - 3):

Coupled mode theory

We use coupled mode theory [1,2,3] to predict the location of the BIC. For simplicity, we consider a reduced two-dimensional coupled rectangular waveguide-resonator system shown in Fig. S 1.

Fig. S 1. Schematic drawing of a coupled two-dimensional waveguide-resonator system.

To make the conclusion as general as possible, we set the width of the waveguide $d = 1$ (unitless), and the width and height of the resonator are L_x and L_y , respectively. Also, the center of the resonator is set as the origin, and the left and right waveguides are attached along the x -axis. Thus, the waveguide spans from $y = -1/2$ to $y = +1/2$. The first step is to compute the eigenfrequencies and eigenmodes of a closed resonator. They eigenfrequencies can be solved analytically with Neumann boundary conditions as follows

$$\frac{\nu_{m,n}^2}{\omega_0^2} = \left(\frac{(m-1)}{L_x} \right)^2 + \left(\frac{(n-1)}{L_y} \right)^2, \quad n, m = 1, 2, 3, \dots \quad (2)$$

where $\nu_{m,n}$ is the resonant frequency and $\omega_0 = \pi c/d$, c is the speed of sound in air. We obtain the corresponding modes ψ by

$$\psi_{m,n} = \sqrt{\frac{(2 - \delta_m^1)(2 - \delta_n^1)}{L_x L_y}} \cos\left(\frac{\pi(m-1)(2x + L_x)}{2L_x}\right) \cos\left(\frac{\pi(n-1)(2y + L_y)}{2L_y}\right) \quad (3)$$

with δ_n^1 and δ_m^1 being the Kronecker delta. The propagating wave numbers in the waveguide are given by

$$\frac{\nu^2}{\omega_0^2} = \frac{(k_p^2)}{\pi^2} + (p-1)^2 \quad (4)$$

with k_p being the wavenumber of the p th channel of the waveguide. We obtain the corresponding modes ϕ by

$$\phi_p = \sqrt{(2 - \delta_p^1)} \cos\left(\frac{\pi(p-1)(2y + 1)}{2}\right) e^{ik_p x}. \quad (5)$$

Then the coupling matrix elements between eigenmodes of closed resonator and p th propagation channels of the left/right waveguide can be obtained by

$$W_{m,n;p} = \int_{-\frac{1}{2}}^{\frac{1}{2}} \psi_{m,n}(x = -\frac{L_x}{2}, y) \phi_p(x = -\frac{L_x}{2}, y) dy. \quad (6)$$

After obtaining the coupling matrix, we compute the complex eigenvalues of the effective Hamiltonian [4,5,6,7], where the real parts correspond to the resonance frequencies and the imaginary parts to the half resonance linewidth. Thus, the search for BICs amounts to finding the zero imaginary part of the eigenvalues. In general, the eigenfunction of any BIC can be decomposed as

$$\phi_{BIC} = \sum_{m,n} a_{m,n} \psi_{m,n}(x, y). \quad (7)$$

Since the BIC is perfectly decoupled from the continuum, its eigenfunction must be given by

$$\int_{-\frac{1}{2}}^{\frac{1}{2}} \phi_{BIC}(x = -\frac{L_x}{2}, y) dy = 0. \quad (8)$$

When two resonant states approach each other as a function of a certain continuous parameter, interference causes an avoided crossing of the two states in their energy positions. At the same time, one of the resonance line widths vanishes exactly at a certain value of the parameter and the other one is boosted to maximum. This is known as Friedrich-Wintgen BIC [8]. Typically, a pair of eigenmodes M_{mn} and $M_{m+2,n-2}$ (or M_{mn} and $M_{m-2,n+2}$) is often used to construct Friedrich-Wintgen BICs. The essence of finding Friedrich-Wintgen BICs is to find two degenerate resonances in a closed resonator with a certain size ratio.

In the present work, we consider the Friedrich-Wintgen BIC in a rectangular resonator embedded in the first channel $p = 1$, provided that other channels are closed for $\nu < 1$. There are numerous degeneracies in a closed rectangular resonator

$$\frac{m^2}{L_x^2} + \frac{n^2}{L_y^2} = \frac{m'^2}{L_x^2} + \frac{n'^2}{L_y^2}. \quad (9)$$

The lowest case corresponds to $m, n = 1, 3$ and $m', n' = 3, 1$ for a square resonator $L_x = L_y$.

After the introduction of the left and right waveguides, these two modes M_{13} and M_{31} are strongly coupled to each other, giving rise to an increase in the destructive interference at a given size ratio. Thus, the resonance frequencies of two modes experience avoided crossing. At the same time, one of the imaginary parts is suppressed to zero while the other is boosted to maximum. Therefore, the formation of such a BIC can be mainly attributed to the destructive interference of modes M_{13} and M_{31} in a closed resonator. We can approximate the eigenfunction of this Friedrich-Wintgen BIC as a superposition of the two eigenmodes of the closed resonator, and its coefficients A and B can be rigorously calculated by

$$\psi_{BIC}(x, y) \approx A\psi_{31}(x, y) + B\psi_{13}(x, y). \quad (10)$$

Substituting Eq. (10) in Eq. (8) gives us

$$A = -W_{1,3;p=1} = -\frac{1}{2\pi} \sqrt{\frac{2L_y}{L_x}} \left[\sin\left(\frac{\pi(Ly+1)}{L_y}\right) - \sin\left(\frac{\pi(Ly-1)}{L_y}\right) \right], \quad (11)$$

$$B = W_{3,1;p=1} = \sqrt{\frac{2}{L_x L_y}}. \quad (12)$$

We can rewrite Eq. (10) as

$$\psi_{BIC}(x, y) \approx \cos\theta\psi_{31}(x, y) + \sin\theta\psi_{13}(x, y), \quad (13)$$

$$\cos\theta = \frac{A}{\sqrt{A^2 + B^2}}, \quad \sin\theta = \frac{B}{\sqrt{A^2 + B^2}} \quad (14)$$

Excellent agreement is found between the eigenfield profile predicted from Eqs.(8-9) and the numerically calculated eigenfield profile of Friedrich-Wintgen BIC, see Fig. S 2.

Fig. S 2. Eigenfield profile. Decomposition of Friedrich-Wintgen BIC into eigenmodes M_{31} and M_{13} .

2. As far as I know, it is not the first time to indirectly measure the acoustic pressure in air by the laser Doppler vibrometry. On the other hand, for the model size in this article, it is possible to directly measure the pressure field by such as 1/4" microphone.

We have incorporated the answer to this question into the answer to your first question. Microphones would not work properly because we are perturbing the high-Q mode, and any energy extraction or backscattering will degrade the QBIC.

3. Could authors explain further the black dotted line in the Fig.1(b)(c)? How do authors calculate them through the coupled mode theory?

Please see the answer to your first comment for a complete theory analysis.

4. Does the technique involved in this article using laser Doppler vibrometry require the structural size of the sample? When it comes to practicality, does this technique work on smaller structures?

We have added an explanation of the physical principle of the LDV (section: Results, Visualization of QBIC, page: 4):

To conduct the refracto-vibrometry, a laser Doppler scanning vibrometer PSV-500 from Polytec¹ is used to measure the changes of the refractive index of the fluid, which is proportional to the acoustic pressure variation within the cavity [43-46]:

$$v(\omega) = \omega \frac{1}{\gamma p_0} \frac{n_0 - 1}{n_0} \int_L p(l, \omega) dl \quad (15)$$

where ω is the angular frequency, v is the LDV velocity, p is the sound pressure, n_0 is the refractive index of air at standard atmospheric pressure, p_0 is the static atmospheric pressure, and γ is the specific heat capacity ratio of air. The basic principle of the LDV is based on the well-known Doppler shift. The pressure waves inside the cavity slightly shift the phase of the emitted monochromatic laser light. The superimposition of the reflected and emitted laser light produces a speckle pattern on the photodetector, which allows the measurement of the corresponding frequency of the pressure waves and the change in the refractive index. The latter is proportional to the sound pressure inside the cavity. This makes it possible to visualize the corresponding pressure distribution.

We have also added a more detailed explanation of the visualization and its limits (section: Methods, pages: 9 and 10):

Visualization. We use refracto-vibrometry to visualize the sound pressure field inside the cavity. A laser Doppler scanning vibrometer PSV-500 from Polytec is used to measure the changes in the refraction index of the fluid, which is proportional to the acoustic pressure variation within the cavity. Overall, 225 measurement points were sequentially recorded with a sampling frequency of 50 kHz for a duration of 2 ms, while the measurements were triggered by the sinusoidal sound generator. To ensure an optimal signal-to-noise ratio, a highly reflective sheet was placed against the rigid surface behind the sample to improve diffuse light reflection. Note that usually a LDV is used for surface normal vibration measurements of structures but captures the pressure wave-induced variation in the refraction index when measured against a rigid surface. In the case of a low-vibration surface, the velocity measurement from the LDV is dominated by the dynamic phase, caused by the sound pressure fluctuations, and the changed refractive index of the acoustic medium, along the traveling path of the light. To ensure the required rigidity, a single point LDV (Polytec PDV-100) measured the surface vibration of the rigid surface from the opposite direction. The surface velocities were found to be orders of magnitude smaller than the signal of the scanning PSV500, confirming that the acoustic pressure dominates the measured results. As the LDV works up to frequencies of 1 MHz, the frequency range is not a limiting factor for pressure field mapping. The size of the structure can be much smaller than those presented in this article and is only limited by the focal point of the laser (25 μm). We use such large structures here because we need to measure the transmission/reflection spectra using an impedance tube with a diameter of 40 mm. Since the Helmholtz equation is linearly scalable, our results can be transferred to different frequency ranges.

¹Polytec GmbH, Waldbronn, Germany

Reviewers' Comments:

Reviewer #1:

Remarks to the Author:

After a thorough examination of the authors' response, I have found that the current version exhibits higher quality. Consequently, I recommend accepting it in its current form.

Reviewer #2:

Remarks to the Author:

The authors have carefully taken account of the comments and added several improvements to the manuscript. So, I would recommend the acceptance of the paper for publication

Reviewer #3:

Remarks to the Author:

The authors have revised the paper as required.